# Low-value MRI of the knee in Norway: a register-based study to identify the proportion of potentially low-value MRIs and estimate the related costs

Bjørn Hofmann ,[1,2] Vegard Håvik,[3] Eivind Richter Andersen,[1] Ingrid Øfsti Brandsæter,[1] Elin Kjelle [1]

¹Department of Health Sciences in Gjøvik, Norwegian University of Science and Technology, Faculty of Medicine and Health Sciences, Gjøvik, Norway
²Centre of Medical Ethics, University of Oslo, Norway, Oslo, Norway
³Department for Medical Reimbursement, Norwegian Directorate of Health, Oslo, Norway

**Correspondence to**
Dr Bjørn Hofmann;
bjoern.hofmann@ntnu.no

## ABSTRACT

**Objectives** The objective of this study is to investigate the proportion of potentially low-value knee MRI in Norway and to provide an estimate of the related costs.

**Design** Register study based on conditional data extraction and analysis of data from Control and Reimbursement of Healthcare Claims registry in Norway.

**Setting** MRI in public specialist healthcare with universal health coverage (Norway).

**Participants** 48 212 MRIs for 41 456 unique patients and 45 946 reimbursement claims.

**Outcome measures** Proportion of MRIs of the knee that (1) did not have a relevant tentative diagnosis prior to the knee MRI, (2) did not have a relevant alternative image of the knee before the MRI or (3) did not have a relevant code from the specialist care within 6 months after the MRI, and those that had combinations of 1, 2 and 3. Estimated costs for those that had combinations of 1, 2 and 3.

**Results** Very few patients (6.4%) had a relevant diagnosis code or prior imaging examination when having the MRI and only 14.6% got a knee-related diagnosis code from the specialist care within 6 months after the MRI. 21.8% of the patients had knee X-ray, CT or ultrasound within 6 months before the MRI. Between 58% and 85% of patients having knee MRIs in Norway have no relevant examinations or diagnoses six months prior to or after the MRI examination. These examinations are unlikely to benefit patients and they correspond to between 24 108 and 35 416 MRIs at a cost of €6.7–€9.8 million per year.

**Conclusion** A substantial proportion of MRIs of the knee in Norway have no relevant examinations or diagnoses before or after the MRI and are potentially of low value. Reducing low-value MRIs could free resources for high-value imaging, reduce waiting times, improve the quality of care and increase patient safety and professional integrity.

## INTRODUCTION

Radiological examinations are invaluable in modern patient care, as they facilitate accurate diagnostics, expedite crucial treatment and reduce morbidity and mortality.[1] On the other hand, uncritical and inappropriate use of radiology may reduce quality and safety of care, for example, in terms of false

### STRENGTHS AND LIMITATIONS OF THIS STUDY

⇒ This study is unique in providing estimates of the proportion and costs of low-value MRI imaging of the knee for a whole nation.
⇒ High-quality registers and uniform coding practices provide reliable overall estimates.
⇒ The results are specific to a public healthcare setting with universal coverage.
⇒ The study is based on aggregate data and not of detailed analyses of individual cases.
⇒ Estimates are not true values as they are based on several specific assumptions.

(positive and negative) results, misdiagnosis, incidental findings of uncertain significance, overdiagnosis, delays in appropriate care and increased costs.[2 3] Accordingly, it is crucial to reduce the use of low-value examinations, that is, examinations that do not lead to change in patient management or are deemed not to be beneficial.[4]

MRI of the knee for knee pain without red flags (eg, without mechanical symptoms or effusion, unless the patient has not improved after completion of a rehabilitation programme) is recognised as low value by the international Choosing Wisely campaign[5 6] and is the second most commonly studied musculoskeletal low-value examination worldwide.[7] Eight studies reported a change in treatment in less than 1% of the cases after an MRI of the knee.[8–15] Further, imaging practice does not always concur with other health-related practices. While there has been a reduction in knee arthroscopy for specific conditions and age groups, the number of knee MRIs has increased substantially, and a recent study from Australia reported a sevenfold increase from 2003 to 2017.[13] While some of these MRIs might have influenced patient management decisions, such as shifting from arthroscopy to arthroplasty, it remains questionable

whether such a volume of MRIs significantly improves patient care.

Accordingly, a large proportion of low-value knee MRIs is an indicator of poor-quality care, waste of resources and a cause of unnecessary waiting times and delayed diagnosis and treatment. We, therefore, need more knowledge about the proportion and costs of this barrier to effective and efficient care. This is crucial for identifying and targeting the problem and for directing health policy-making (deimplementation, disinvestment).

Many studies document geographical variations in the use of knee MRIs.[16 17] While such variations indicate both underuse and overuse of services,[18 19] they do not indicate the 'right level' or what is of low and high value. We, therefore, need more information on which examinations that are (not) valuable for patients, health providers and the healthcare system. While detailed analyses of individual patients' pathways can provide insights into the appropriateness of single examinations, this level of analysis is unfeasible on a national or aggregated level. Moreover, cost from unnecessary MRIs from single consultants in the National Health Service has been estimated,[20] but we need aggregated national estimates.

Consequently, there is a need to document both how many of the MRIs of the knee that potentially are of low value and what the costs are. Therefore, by using register data to identify patients who did not have relevant diagnoses or actions 6 months before or after an MRI of the knee, the objective of this study is to estimate the proportion of examinations that are unlikely to have benefited patients and that potentially are of low value and to provide an estimate of the related costs.

The research questions are as follows: (1) How large is the proportion of the MRIs of the knee that do not have relevant examinations or diagnoses 6 months before or diagnoses after the examination in Norway and (2) What are the related costs per year for these examinations?

## METHODS
### Setting
In Norway, all residents are entitled to essential medical care,[21] and due to universal health coverage, most health services are publicly funded through taxes.[22] Specialists and radiological services are organised within the specialist care, governed by the regional hospital trusts.[23] Physiotherapist and general practitioners (GP) are organised in the primary care which are the municipalities' responsibility. However, several orthopedist and physiotherapist clinics operate in the private healthcare sector either on contracts with the public health system or independently.[24]

Private imaging centres perform about 20%–25% of all outpatient examinations, where 60% of these are CT or MRI examinations.[25] A significant portion of these examinations is commissioned by public health services to ease the pressure on public services.[23] Further, private imaging centres offer insurance or out-of-pocket

financed radiology, which allows shorter waiting times for patients. About 10% of the population have private health insurances.[22]

### Data
This study uses data from Control and Reimbursement of Healthcare Claims (CRHC) registry by the Norwegian Directorate of Health. Specified types of registry data were retrieved for all patients who performed an outpatient knee MRI at public hospitals and/or private imaging centres during a 6-month period (1 July 2021 to 31 December 2021). Only publicly employed healthcare providers or providers with contracts with the public health system report to CRHC. Independent orthopaedists or physiotherapists do not report to CRHC and are thus not included in this study.

Both hospitals and private imaging centres report radiological examinations to CRHC by using the Norwegian Classification of Radiological Procedures (NCRP) codes. In this study, the NCRP code for knee, MRI-SNG0AG, was used to extract relevant procedures.

### Analysis
As it is not feasible on a national level to analyse the medical history, indications and clinical pathway of all patients, we assume that the MRIs of the knee that do not have relevant diagnoses or actions 6 months before or after the examination are unlikely to have benefited the patient and are potentially low-value examinations.

Moreover, the analyses are based on the assumptions that (1) the MRI of the knee is appropriate and of high value if the patient has received a relevant diagnostic code before the MRI, (2) the MRI has been of value to the patient if the patient has a knee-related diagnostic code after the examination, (3) patients undergoing other imaging examinations prior to the MRI experience value from the additional MRI, (4) 10% of patients migrate to private treatment of the knee and are not identified in the public health registries and (5) imaging in patients with a tentative knee diagnosis that do not lead to any active interventions (do nothing), are considered to be of high value as the pathways of the patients have changed.

Consequently, to estimate the proportion of knee MRIs that potentially are of low value, we identified all patients who (1) did not have a relevant tentative diagnosis prior to the knee MRI, (2) did not have a relevant alternative image of the knee before the MRI or (3) did not have a relevant code from the specialist care within 6 months after the MRI. We then investigated combinations of 1–3. The logic model of the data retrieval is illustrated in figure 1.

To identify the potentially valuable MRIs, we used diagnosis codes reported from GPs and physiotherapists (ICPC-2 codes: L78 and L96) within 6 months prior to the MRI and diagnosis codes reported from orthopaedic clinics recorded within 6 months after the MRI was performed (ICD-10 codes: M22, M23, M66.0, M70.5, M71.2 and M92.4). Moreover, we identified patients

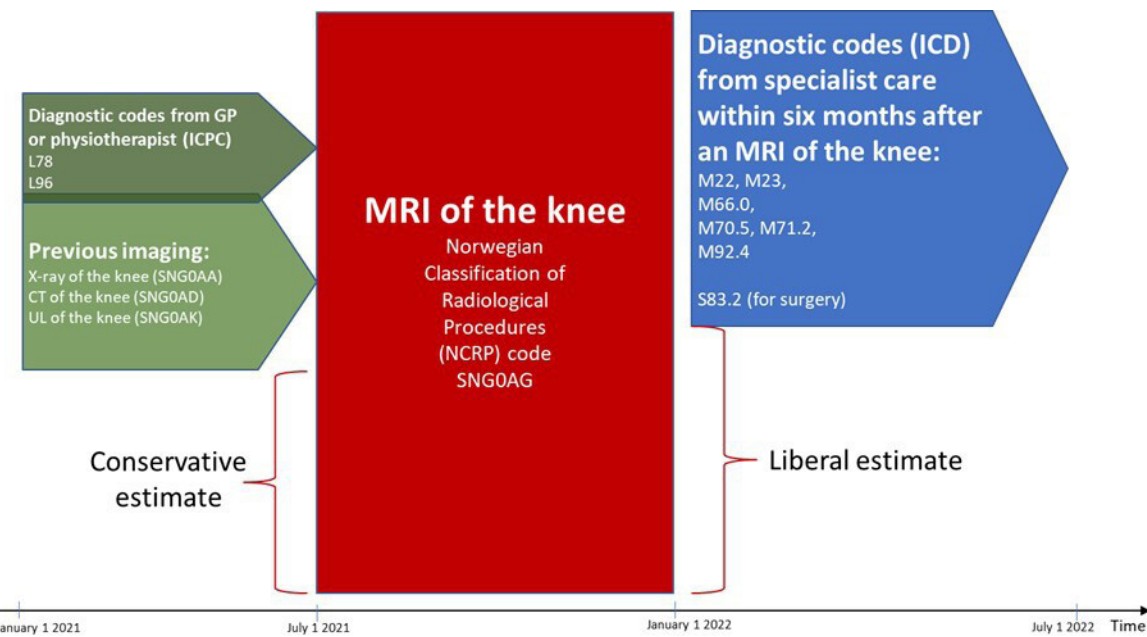

**Figure 1** Illustration of the logic model of the data retrieval and indication of how two of the estimates of the proportion of low-value imaging were calculated. Diagnostic codes from ICPC-2 and other examinations 6 months prior to the MRI of the knee make the examination count as high value. The same goes for listed ICD-codes within 6 months after the examination. The liberal estimate is based on the proportion of MRIs without healthcare-relevant implications of the imaging within 6 months after the MRI. The conservative estimate is based on the proportion of MRIs where patients did neither have a relevant code from their GP or physiotherapist or relevant imaging codes within 6 months prior to the MRI and nor a relevant code from the specialist care within 6 months after the MRI. GP, general practitioner. ICD = International Classification of Diseases (WHO). ICPC-2 = International Classification of Primary Care, 2nd edition (WHO).

who had an X-ray, CT or ultrasound of the knee within 6 months prior to the MRI, as we assumed these patients would have some value from the MRI, for example, when referred to specialist care. These examinations were identified through NCRP codes: SNG0AA, SNG0AG and SNG0AK.

Correspondingly, we analysed the number of patients who did neither have relevant imaging codes within 6 months prior to the MRI nor have any of the relevant codes from the specialist care within 6 months after the MRI, and therefore, excluded MRIs which could be deemed necessary by specialists.

Additionally, we investigated the number of patients who did not have a relevant code from their GP or physiotherapist or relevant imaging codes within 6 months prior to the MRI and did not have a relevant code from the specialist care within 6 months after the MRI.

An overview of the radiological and diagnosis codes is given in table 1.

Cost estimates were based on public cost per patient data for MRI in Norway.

Conditional data extraction and analysis is done with SQL developer V.21.4.1.349. The queries made by Department for Medical Reimbursement, Norwegian Directorate of Health, are provided in online supplemental file 1.

### Patient and public involvement
Representatives of the Norwegian patient organisation and an expert committee with representatives from the involved professions and authorities were involved on the research project from which this study stems.

### RESULTS
A total of 48 212 MRIs were conducted during the period, divided by 41 456 unique patients and 45 946 reimbursement claims (used for categorising reimbursements from CRHC).

A total of 33 234 (80.2%) patients did neither have relevant codes (L78, L96) from their GP or physiotherapist within a period of 6 months prior to the MRI nor have any of the relevant diagnosis codes from the specialist care within 6 months after the MRI.

A total of 28 140 (67.9%) patients did neither have relevant imaging codes (SNG0AA, SNG0AD, SNG0AK) within 6 months prior to the MRI nor have any of the relevant codes from the specialist care in table 1 within 6 months after the MRI.

A total of 26 786 (64.6%) patients did not have a relevant code from their GP or physiotherapist or relevant imaging codes within 6 months prior to the MRI and did not have a relevant code from the specialist care within 6 months after the MRI.

Moreover, we found that 6040 (14.6%) patients got a relevant diagnosis code for the knee within the first 6 months after the MRI. 2201 (5.3%) patients had a relevant diagnosis code for the knee from their GP prior to the MRI, and 461 (1.1%) patients had relevant codes from a

**Table 1** Description of radiological and diagnosis codes and reason why these codes are used to identify appropriate MRI examinations

| Code | Description | Reason for subtracting patients with this code |
|------|-------------|------------------------------------------------|
| **Codes registered within 6 months prior to MRI** | | |
| NCRP-SNG0AA | X-ray of the knee | Findings or lack of findings on these examinations could indicate an MRI is appropriate |
| NCRP-SNG0AG | CT of the knee | |
| NCRP-SNG0AK | Ultrasound of the knee | |
| ICPC-2 L78 | Distortion of the knee | MRI for these conditions could be considered appropriate |
| ICPC-2 L96 | Acute inner knee damage | |
| **Codes registered within 6 months after MRI** | | |
| ICD-10 M22 | Disorders of patella | MRI for these conditions could be considered appropriate |
| ICD-10 M23 | Internal derangement of knee | |
| ICD-10 M66.0 | Rupture of popliteal cyst | |
| ICD-10 M70.5 | Other bursitis of knee | |
| ICD-10 M71.2 | Synovial cyst of popliteal space (Baker) | |
| ICD-10 S83.2 | Tear of meniscus, current | |
| ICD-10 M92.4 | Juvenile osteochondrosis of patella | |

ICD-10, International Classification of Diseases (WHO), 10th edition. ; ICPC-2, International Classification of Primary Care, 2nd edition (WHO); NCRP, Norwegian Classification of Radiological Procedures.

publicly registered physiotherapist. 9016 (21.8%) patients had knee X-ray, CT or ultrasound within 6 months before the MRI. Figure 2 provides an overview of these results.

In 2021, the total number of outpatient MRIs of the knee were 87 140. Mean cost per patient for knee MRI in 2021 at public hospitals was €362 (US$428), and at private imaging centres €256 (US$303). In 2021, approximately 80% of outpatient MRIs if the knee were conducted at different private imaging centres. Based on this distribution, the total number of knee MRI amounts to a cost of €23.8 million (US$28.2 million).

## Interpretation

The MRIs of the knee that do not have relevant diagnoses or actions before or after the MRI and that these are not likely to be important to patients' health and are potentially of low value. However, the estimates of the proportion of low-value imaging depend on how you define low-value imaging.

If all cases of MRI for persons who have no healthcare-relevant implications of the imaging within 6 months after the MRI count as low value, 85.43% ((41 456–6040)/41 456) of all patients had a low-value MRI.

If patients who did neither have relevant imaging codes within 6 months prior to the MRI nor have any of the relevant codes from the specialist care within 6 months after the MRI count as low value, then 67.88% (28 140/41 456) of all patients had a low-value MRI.

Moreover, if those cases where patients who did not have a relevant code from their GP or physiotherapist or relevant imaging codes within 6 months prior to the MRI and did not have a relevant code from the specialist care within 6 months after the MRI count as low value, then 64.61% (26 786/41 456) of all patients had a low-value MRI.

Additionally, if the same percentage of those without any codes within 6 months before or after the MRI had private insurance as the general population in Norway (10%), and all of these did receive healthcare services from private providers not reporting to CRHC (n=2678), and if none of these were of low value, then 58.15% of all patients had a low-value MRI.

Table 2 provides an overview of the various estimates of the proportion of low-value MRIs and the corresponding yearly costs.

## DISCUSSION

This analysis of registry data showed that very few patients (6.4%) had a relevant diagnosis or imaging code registered prior to an outpatient MRI, and that 14.6% got a knee-related diagnosis code from the specialist care within 6 months after the MRI. Between 58% and 85% of the MRIs of the knee are estimated to be of low value at a cost of €6.7–€9.8 million (US$7.9–US$11.6 million).

The findings are in line with the international literature showing that 40%–95% of knee MRIs were inappropriate or unnecessary.[8–15] The results (85% low-value MRIs of the knee) are also in line with the results of an internal unpublished report at one of the major hospitals in Norway (2005).

In 2004, the rate of knee MRIs was 156,[26] while the average number increased to 173 for the period 2012–2015.[23] Interestingly, the overall annual rate of arthroscopic procedures declined by 33% from 2012 to 2016.[27] Further, geographical variations in knee MRIs conducted in Norway are well documented, ranging from 117 to 211 examinations per 10 000 inhabitants.[23]

Accordingly, the findings fit with the documented discrepancy between imaging use and outcome.[5–7] Extensive use of imaging with low direct outcomes may indicate that patients and clinicians acquire imaging without a plan for using the results on a nice-to-know basis.[28] Unfortunately, our findings do not falsify this hypothesis as only about 6% of the patients had a tentative diagnosis code

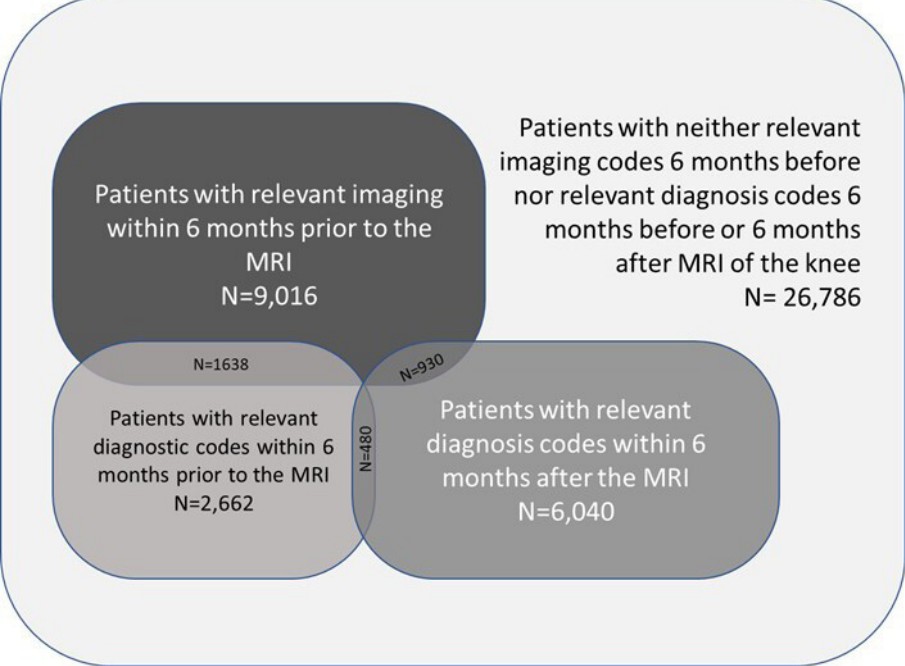

**Figure 2** Illustration of the relationship between the various identified groups: patients with relevant diagnostic codes 6 months prior to MRI of the knee; patients with other relevant imaging 6 months prior to MRI of the knee; patients with relevant diagnostic codes within 6 months after the MRI of the knee; patients with neither of the above-mentioned qualifications. Overlaps are shown in number of images.

for the knee from their GP or physiotherapist prior to the MRI examination.

Disturbingly, the results may also indicate a practice where doctors are afraid of giving a diagnosis based on clinical examinations without imaging, that is, lowering clinical confidence and competency and weakening the gate-keeper function. According to the Norwegian guidelines for musculoskeletal imaging explicitly state that clinical assessment is the primary diagnostic tool.[29]

Interestingly, only about 22% of the patients had another type of knee imaging prior to the MRI. This finding indicates that advanced imaging (MRI) is referrers' primary choice of imaging. However, for a common specific conditions, knee osteoarthritis, the primary imaging modality is X-ray, not MRI.[29] This adds to suggested practice change in the literature on low-value imaging, proposing taking X-ray before conducting knee MRI as a warranted way to optimise the use of imaging examinations.

Our most conservative estimate shows that more than 58% of MRI are potentially of low value, but the real number of low-value knee MRIs is likely higher. International studies show that less than 1% of knee MRI results yielded change in patient management.[7] Thus, the potential for shifting resources from low-value MRIs to high-value MRIs in the Norwegian healthcare system in a safe manner is substantial.

### Limitations

In this study, we have made a series of assumptions. First, we assumed that patients followed up in the specialist healthcare system with relevant knee-related (ICD) codes within 6 months after the MRI would have some value from the examination. This assumption may not hold, as the diagnoses may have been set from other examinations than the MRI (or the patient may have had a knee accident during this period). If so, our estimates of low-value

| Table 2 | Overview of the estimates of the proportion of low-value MRIs and the corresponding yearly costs | | | |
|---|---|---|---|---|
| **Estimate** | **Most conservative** | **Conservative** | **Moderate** | **Liberal** |
| Proportion (%) | 58.15 | 64.61 | 67.88 | 85.43 |
| Number of potentially low value MRIs | 24 108 | 26 786 | 28 140 | 35 416 |
| Estimation | =(26 786–2678)/41 456 | =26 786/41 456 | =28 140/41 456 | =(41 456–6040)/41 456 |
| Cost estimates in million EUR, (USD) | 6.7 (7.9) | 7.4 (8.8) | 7.8 (9.2) | 9.8 (11.6) |

MRIs would be too low. On the other hand, diagnoses from the specialist healthcare may have been set more than 6 months after the MRI, resulting in overestimating low-value imaging. However, waiting times are much shorter than 6 months (especially in private institutions where most of these examinations are performe), so this may be unlikely.

Second, we assumed that an MRI would be indicated for patients with relevant diagnostic codes related to the knee from their GP or physiotherapist within 6 months prior to the MRI. This may not hold, as diagnoses may be set to provide reasons for having an MRI (eg, if requested by the patient).[30]

Third, we also assumed that patients who had an X-ray, CT or ultrasound of their knee within 6 months prior to the MRI would have some value from the examination. This may not always be the case, making us underestimate the proportion of low-value imaging.

Fourth, our data did not include data from independent orthopaedists and physiotherapists. This would lead to an over estimation of low-value MRIs as some patients might be treated for relevant diagnosis in the private health services. To compensate for this, we assumed a 10% rate of patients treated in private independent healthcare and that all of these are of high value. However, many of these enter back into the public healthcare (with new MRIs) after private treatment and are included in our estimates. Hence, the most conservative estimate may provide too low estimates.

Fifth, some examinations may have 'do nothing' as an outcome, for example, as an alternative to an active intervention. This would mimic a low-value examination in our study. However, if 'do nothing' follow a tentative diagnosis before the MRI, they do not count as low-value care, as the care pathway is changed.

Final, data from in-patient examinations or examinations paid out-of-pocket were not included. The number of in-patient examinations of knee MRIs is very low, and the out-of-pocket examinations are not relevant in a study on the public healthcare system with a societal perspective.

Another source of error may be that there are other codes that are relevant than the ones we have identified, or that clinical conditions are under-reported. To reduce the first problem, we have discussed the codes with experts in the field, and the codes have been suggested by specialists. The second source of error may be small as there is a high coding consciousness in Norway as reimbursements depend on reporting of codes.[31] While the clinical information about patients is limited in this study, the model used for identifying appropriateness is assessed as relevant to overcome this challenge in identifying potential low-value knee MRIs. From related areas we know that coding may be inaccurate,[32] however, we have no knowledge about this source of poor performance.

The study was conducted just after most restrictions of the COVID-19 pandemic (2020–2021) were lifted and a backlog of MRIs could be expected. However, this should have resulted in a higher number of high-value MRIs of the knee. As seen from another study, high-value imaging did not have specific priority during the pandemic.[33]

It is important to note that this study is performed in a public healthcare system with universal coverage. The premises, measures and outcomes may be different in other settings.

## CONCLUSION

Very few patients (6.4%) had a relevant diagnosis code or prior imaging examination when having the MRI and only 14.6% got a knee-related diagnosis code from the specialist care within 6 months after the MRI. 21.8% of the patients had knee X-ray, CT or ultrasound within 6 months before the MRI.

This indicates that between 58% and 85% of patients having knee MRIs in Norway have potentially low-value imaging, as they have no relevant diagnoses or actions prior or after the MRI examination.

The total number of low-value MRIs of the knee per year would be between 24 108 and 35 416, and amount to a cost of €6.7–€9.8 million for 2021.

A substantial part of MRIs of the knee in Norway is potentially of low value. Reducing low-value MRIs could free resources for high-value imaging, reduce waiting times, improve the quality of care, and increase patient safety and professional integrity.

**Acknowledgements** We are most thankful to the Directorate of Health for their help with the analyses.

**Contributors** BH designed the study with input from VH, ERA, IØB and EK. VH performed the data analyses. BH wrote the first draft of the manuscript and all authors provided substantial input for several rounds of subsequent revisions. BH acts as guarantor for the work.

**Funding** This project received financial support from the Research Council of Norway (project number 302503).

**Competing interests** None declared.

**Patient and public involvement** Patients and/or the public were not involved in the design, or conduct, or reporting, or dissemination plans of this research.

**Patient consent for publication** Not applicable.

**Provenance and peer review** Not commissioned; externally peer reviewed.

**Data availability statement** All data relevant to the study are included in the article or uploaded as online supplemental information.

**ORCID iDs**
Bjørn Hofmann http://orcid.org/0000-0001-6709-4265
Elin Kjelle http://orcid.org/0000-0001-6370-2729

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
