## [Reviewer comments · BMJ Open]

ARTICLE DETAILS

TITLE (PROVISIONAL)	Potentially low value MRI of the knee in Norway: a register-based study to identify the proportion of potentially low-value MRIs and estimate the related costs
AUTHORS	Hofmann , Bjørn; Håvik, Vegard; Andersen, Eivind; Brandsæter, Ingrid; Kjelle, Elin

VERSION 1 – REVIEW

REVIEWER	Wei, Shijun Wuhan General Hospital of Guangzhou Command, Orthopaedics
REVIEW RETURNED	26-Dec-2023

GENERAL COMMENTS	In this manuscript, the authors investigated the proportion MRI imaging of the knee that are of low value to patients in Norway. Based on this, they further estimated the related costs of low-value MRI of the knee. This work provides new insight and opinion into the MRI examinations. Reducing low-value MRIs is highly meaningful, which could free resources for high-value imaging, reduce waiting times, improve the quality of care and so on. The manuscript is well-organized and stated. I would suggest accepting it after the following concerns are addressed. 1. In this study, the authors listed a few factors that could affect the accuracy of the results. Among the data acquired, the authors need to clearly state how to exclude the proportion of MRI imaging of knee that could be high value for patients, especially when MRI examination is deemed necessary by specialists.2. In the Figure 1 and Figure 2, the logic model of the data retrieval, also the relationship between various results were not clearly stated, detailed information or illustrations should be added in each figure legend.
--

REVIEWER	Karatekin, Yavuz Selim Ministry of Health Samsun Education and Research Hospital, Orthopedics and Traumatology
REVIEW RETURNED	25-Jan-2024

GENERAL COMMENTS	I congratulate the researchers for their efforts. However, I am of the opinion that the methodology in the planning of the article is not appropriate. Relying solely on ICD codes for data definition in low-value MRI imaging may result in negative conclusions, defining only through codes and potentially leading to an overestimation of unnecessary MRI imaging. Nevertheless, a clinical study, accompanied by experts and based on sampling from the relevant population, could determine the appropriateness of MRI
--

	requirements. Therefore, I believe that such a study will not reach accurate conclusions without contacting the patients, conducting individual analyses of MRI reports and images, and performing a careful examination ** Background Page3, line 22. What are the red flag findings for knee MRI imaging? I believe that giving brief information about this will enrich the article. Page 3 line 26. Studies have generally been conducted for older patients, and in most of these studies, there seems to be no requirement for expert involvement. However, I believe it is not accurate to state definitively that these studies apply to all patients. As indicated in the cited sources, it is essential to add qualifiers such as "in older patients" or "in specific conditions (such as osteoarthritis)" to the sentence. Page 3, line 27-28-29. The cited source (11) indicates a decrease in knee arthroscopy procedures, despite an increase in the number of knee arthroplasties correlating with knee MRI imaging. In earlier years, diagnostic arthroscopy was commonly performed on the knee; however, in contemporary practice, diagnostic arthroscopy procedures are nearly obsolete. One of the primary reasons for this shift is attributed to advancements in MRI imaging. Following the diagnosis of cartilage damage through MRI, especially in older patients, it is believed that they may not benefit from arthroscopy, and the option of arthroplasty becomes more suitable. In other words, MRI imaging can alter the treatment options for patients. After MRI imaging, it may be determined that arthroplasty is a more appropriate choice than knee arthroscopy for addressing cartilage lesions. Additionally, it is crucial to note that the existing source (11) is focused on older patients, and this should be explicitly mentioned in the text. Furthermore, I perceive that this source presents views that are not supportive but rather against the arguments in the article. Method How were the ICD code selections made, and based on what criteria were these code selections defined with specific experts? I am of the opinion that the methodology for defining low-value MRI in this study is not conducted accurately. In the analysis section, I believe that the three definitions made are not sufficient to determine low value. There is a possibility that additional codes, beyond those currently defined, have been identified, and treatment based on these codes may have been applied. Furthermore, for the definition of low-value MRI, the evaluation of MRI report results by an expert is essential. Only when the presence of pathology in these reports is correlated with the patient's relevant complaints can a conclusive judgment be reached. I believe that defining low-value MRI solely based on ICD codes without alignment with patient complaints and MRI findings is not accurate
--	---

Reviewer: 1

Dr. Shijun Wei, Wuhan General Hospital of Guangzhou Command

Comments to the Author:

In this manuscript, the authors investigated the proportion MRI imaging of the knee that are of low value to patients in Norway. Based on this, they further estimated the related costs of low-value MRI of the knee. This work provides new insight and opinion into the MRI examinations. Reducing low-value MRIs is highly meaningful, which could free resources for high-value imaging, reduce waiting times, improve the quality of care and so on. The manuscript is well-organized and stated. I would suggest accepting it after the following concerns are addressed.

RESPONSE: We are most thankful for the acknowledgment of our work and for the valuable inputs.

1. In this study, the authors listed a few factors that could affect the accuracy of the results. Among the data acquired, the authors need to clearly state how to exclude the proportion of MRI imaging of knee that could be high value for patients, especially when MRI examination is deemed necessary by specialists.

RESPONSE: The following clarification is provided in the analysis section: "Moreover, we identified patients who had an x-ray, CT, or ultrasound of the knee within six months prior to the MRI, as we assumed these patients would have some value from the MRI, for example when referred to specialist care'.

'Correspondingly, we analysed the number of patients that did neither have relevant imaging codes within six months prior to the MRI nor have any of the relevant codes from the specialist care within six months after the MRI and therefore excluded MRIs which could be deemed necessary by specialists'.

2. In the Figure 1 and Figure 2, the logic model of the data retrieval, also the relationship between various results were not clearly stated, detailed information or illustrations should be added in each figure legend.

RESPONSE: Good point. The legend for Figure 1 now reads: "Illustration of the logic model of the data retrieval and indication of how two of the estimates of the proportion of low value imaging were calculated. Diagnostic codes from ICPC-2 and other examinations six months prior to the MRI of the knee make the examination count as high value. The same goes for listed ICD-codes within six months after the examination. The liberal estimate is based on the proportion of MRIs without healthcare-relevant implications of the imaging within six months after the MRI. The conservative estimate is based on the proportion of MRIs where patients did neither have a relevant code from their GP or physiotherapist or relevant imaging codes within six months prior to the MRI and nor a relevant code from the specialist care within six months after the MRI." For Figure 2 the legend now reads: "Illustration of the relationship between the various identified groups: patients with relevant diagnostic codes six months prior to MRI of the knee; patients with other relevant imaging six months prior to MRI of the knee; patients with relevant diagnostic codes within six months after the MRI of the knee; patients with neither of the above-mentioned qualifications. Overlaps are shown in number of images."

Reviewer: 2

Dr. Yavuz Selim Karatekin, Ministry of Health Samsun Education and Research Hospital

Comments to the Author:

I congratulate the researchers for their efforts. However, I am of the opinion that the methodology in the planning of the article is not appropriate. Relying solely on ICD codes for data definition in low-

value MRI imaging may result in negative conclusions, defining only through codes and potentially leading to an overestimation of unnecessary MRI imaging. Nevertheless, a clinical study, accompanied by experts and based on sampling from the relevant population, could determine the appropriateness of MRI requirements. Therefore, I believe that such a study will not reach accurate conclusions without contacting the patients, conducting individual analyses of MRI reports and images, and performing a careful examination

RESPONSE: We are thankful for this valuable assessment. We agree that the study will not reach accurate conclusions on individual cases. Individual audits and case reviews are necessary for that. However, as health policy and quality improvement strategies need information on overall proportions of low-value imaging and their related costs, aggregate estimates are needed. This study estimates the proportion of MRIs might be of low-value, and therefore indicates areas for improvement where common resources could be used in a better way. Coding practices in Norway are homogenous (especially for this examination) and we have national registries of high quality. Hence, this ascertains that the estimates are as reliable. Moreover, our estimates are conservative in order to avoid overestimation.

To clarify that we do not intend to provide exact analysis, we have changed the introduction, now stating:

'Many studies document geographical variations in the use of knee MRIs,^{14 15} and such variations indicate both underuse and overuse of services.^{16 17} The presence of such variations consequently indicate the existence of poor imaging practice, underscoring the importance of further investigations of the proportion and magnitude of the problem. Moreover, cost from unnecessary MRIs from single consultants in the NHS have been estimated,¹⁸ but we need aggregated national estimates. Consequently, there is a need to document both how many of the MRIs of the knee that are of low value and what the costs are. However, precise evaluations of value necessitate individual scrutiny of MRI reports and subsequent healthcare. This level of analysis is unfeasible on a national or aggregated level. Therefore, by using register data and a number of assumptions, the objective of this study is to investigate the proportion of potentially low-value knee MRI imaging in Norway and to provide an estimate of the related costs. The research questions are: 1) how large is the extension of potentially low-value MRI imaging of the knee in Norway, and 2) what are the related costs?'

Also, the conclusion is amended, now stating: "A substantial part of MRIs of the knee in Norway is potentially of low value.

Background

Page3, line 22.

What are the red flag findings for knee MRI imaging? I believe that giving brief information about this will enrich the article.

RESPONSE: Thank you for this input. A description of Red Flags is added to the text.

Page 3 line 26.

Studies have generally been conducted for older patients, and in most of these studies, there seems to be no requirement for expert involvement. However, I believe it is not accurate to state definitively that these studies apply to all patients. As indicated in the cited sources, it is essential to add qualifiers such as "in older patients" or "in specific conditions (such as osteoarthritis)" to the sentence.

RESPONSE: Thank you for the input. The text is amended accordingly.

Page 3, line 27-28-29.

The cited source (11) indicates a decrease in knee arthroscopy procedures, despite an increase in the number of knee arthroplasties correlating with knee MRI imaging. In earlier years, diagnostic arthroscopy was commonly performed on the knee; however, in contemporary practice, diagnostic arthroscopy procedures are nearly obsolete. One of the primary reasons for this shift is attributed to advancements in MRI imaging. Following the diagnosis of cartilage damage through MRI, especially in older patients, it is believed that they may not benefit from arthroscopy, and the option of arthroplasty becomes more suitable. In other words, MRI imaging can alter the treatment options for patients. After MRI imaging, it may be determined that arthroplasty is a more appropriate choice than knee arthroscopy for addressing cartilage lesions. Additionally, it is crucial to note that the existing source (11) is focused on older patients, and this should be explicitly mentioned in the text. Furthermore, I perceive that this source presents views that are not supportive but rather against the arguments in the article.

RESPONSE: Thank you for your input. The revised text now reads: "While there has been a reduction in knee arthroscopy for specific conditions and age groups, the number of knee MRIs has increased substantially, and a recent study from Australia reported a seven-fold increase from 2003-2017.¹¹ Some of these MRIs might have influenced patient management decisions, such as shifting from arthroscopy to arthroplasty, it remains questionable whether such a volume of MRIs significantly enhances patient care. Accordingly, a large proportion of low-value knee MRIs is an indicator of poor-quality care, waste of resources, and a cause of unnecessary waiting times and delayed diagnosis and treatment. While MRI of the knee is an acknowledged low-value service, we need more knowledge on the extension and costs of this barrier to effective and efficient care."

Method

How were the ICD code selections made, and based on what criteria were these code selections defined with specific experts? I am of the opinion that the methodology for defining low-value MRI in this study is not conducted accurately. In the analysis section, I believe that the three definitions made are not sufficient to determine low value. There is a possibility that additional codes, beyond those currently defined, have been identified, and treatment based on these codes may have been applied. Furthermore, for the definition of low-value MRI, the evaluation of MRI report results by an expert is essential. Only when the presence of pathology in these reports is correlated with the patient's relevant complaints can a conclusive judgment be reached. I believe that defining low-value MRI solely based on ICD codes without alignment with patient complaints and MRI findings is not accurate

RESPONSE: As stated in the manuscript: "Another source of error may be that there are other codes that are relevant than the ones we have identified, or that clinical conditions are under-reported. To reduce the first problem, we have discussed the codes with experts in the field, and the codes have been suggested by specialists. The second source of error may be small as there is a high coding consciousness in Norway as reimbursements depend on reporting of codes." More specifically, we also point out that: "While detailed analyses of individual patients' pathways can provide insights into the appropriateness of single examinations, this level of analysis is unfeasible on a national or aggregated level." We are not studying the appropriateness or health-related value of individual examinations (as this is practically impossible for 48,000 MRIs), but we want to provide overarching estimates based on the (lack of) relevant disease or treatment codes. While the first is important for quality improvement interventions, the latter is crucial for health policy making attention and action. We very much would like to do both, but while the first is not feasible, we believe that doing the next best is an important step in the right direction.